# Testosterone is an endogenous regulator of BAFF and splenic B cell number

Anna S. Wilhelmson [1,2], Marta Lantero Rodriguez [1], Alexandra Stubelius[3,4], Per Fogelstrand[1], Inger Johansson[1], Matthew B. Buechler[5], Steve Lianoglou[5], Varun N. Kapoor[5], Maria E. Johansson[6], Johan B. Fagman [7], Amanda Duhlin[8], Prabhanshu Tripathi[9], Alessandro Camponeschi[10], Bo T. Porse [2], Antonius G. Rolink[11], Hans Nissbrandt[12], Shannon J. Turley[5], Hans Carlsten[3], Inga-Lill Mårtensson[10], Mikael C.I. Karlsson [8] & Åsa Tivesten [1]

Testosterone deficiency in men is associated with increased risk for autoimmunity and increased B cell numbers through unknown mechanisms. Here we show that testosterone regulates the cytokine BAFF, an essential survival factor for B cells. Male mice lacking the androgen receptor have increased splenic B cell numbers, serum BAFF levels and splenic *Baff* mRNA. Testosterone deficiency by castration causes expansion of BAFF-producing fibroblastic reticular cells (FRCs) in spleen, which may be coupled to lower splenic noradrenaline levels in castrated males, as an α-adrenergic agonist decreases splenic FRC number in vitro. Antibody-mediated blockade of the BAFF receptor or treatment with the neurotoxin 6-hydroxydopamine revert the increased splenic B cell numbers induced by castration. Among healthy men, serum BAFF levels are higher in men with low testosterone. Our study uncovers a previously unrecognized regulation of BAFF by testosterone and raises important questions about BAFF in testosterone-mediated protection against autoimmunity.

---

[1] Wallenberg Laboratory for Cardiovascular and Metabolic Research, Institute of Medicine, University of Gothenburg, Sahlgrenska University Hospital, Bruna Stråket 16, SE-413 45 Gothenburg, Sweden. [2] The Finsen Laboratory, Rigshospitalet; Biotech Research and Innovation Centre (BRIC); Novo Nordisk Foundation Center for Stem Cell Biology (DanStem), Faculty of Health Sciences, University of Copenhagen, Ole Maaløesvej 5, DK-2200 Copenhagen N, Denmark. [3] Center for Bone and Arthritis Research (CBAR), Institute of Medicine, University of Gothenburg, Sahlgrenska University Hospital, Vita Stråket 11, SE-413 45 Gothenburg, Sweden. [4] Center of Excellence in Nanomedicine and Engineering, University of California San Diego, 9500 Gilman Dr., La Jolla, CA 92093, USA. [5] Department of Cancer Immunology, Genentech, 1 DNA Way, South San Francisco, CA 94080, USA. [6] Department of Physiology, Institute of Neuroscience and Physiology, University of Gothenburg, Box 432, SE-405 30 Gothenburg, Sweden. [7] Sahlgrenska Cancer Center, Department of Surgery, Institute of Clinical Sciences, University of Gothenburg, Box 100, SE-405 30 Gothenburg, Sweden. [8] Department of Microbiology, Tumor and Cell Biology, Karolinska Institute, SE-171 77 Stockholm, Sweden. [9] Centre for Human Microbial Ecology, Translational Health Science and Technology Institute, NCR Biotech Science Cluster, 3rd Milestone Faridabad–Gurgaon Expressway, Faridabad 121001 Haryana, India. [10] Department of Rheumatology and Inflammation Research, Institute of Medicine, University of Gothenburg, Box 480, SE-405 30 Gothenburg, Sweden. [11] Department of Biomedicine, Developmental and Molecular Immunology, University of Basel, Mattenstrasse 28, 4058 Basel, Switzerland. [12] Department of Pharmacology, Institute of Neuroscience and Physiology, University of Gothenburg, Box 431, SE-405 30 Gothenburg, Sweden. Deceased: Antonius G. Rolink. Correspondence and requests for materials should be addressed to Å.T. (email: asa.tivesten@medic.gu.se)

Sex steroid hormones have profound effects on the immune system, and insight into these effects may provide important clues to the sexual dimorphism of immune-dependent disorders. Many autoimmune diseases, such as rheumatoid arthritis and systemic lupus erythematosus (SLE), are less prevalent in men[1] and data suggest that testosterone, the main androgen, may protect against autoimmune disease[1,2]. Androgen deficiency, resulting from various causes such as hypopituitarism or Klinefelter´s syndrome, has been associated with increased risk of female-predominant autoimmune diseases; the risk of SLE increases 18-fold in Klinefelter patients and clinical remission has been reported after testosterone substitution[3]. Testosterone deficiency induced by castration also increases disease activity in mouse models of autoimmune disease such as experimental autoimmune glomerulonephritis and lupus[4,5], and androgen treatment improves survival in male lupus NZB/NZW F1 mice[6].

While the complex effects of oestrogens on adaptive immunity have been extensively studied[7], less is known about how androgens modulate the immune system[8]. Patients with both hypogonadotropic hypogonadism and Klinefelter´s syndrome have higher blood B cell count, which is lowered by testosterone replacement therapy[9,10]. Testosterone suppresses B lymphopoiesis in the bone marrow[8], and we have shown that male general androgen receptor (AR; the receptor for testosterone) knockout (G-ARKO) mice have increased numbers of bone marrow B cell precursors from the pro-B stage[11]. Through studies of osteoblast-lineage cell-specific ARKO (O-ARKO) mice, we also could show that the osteoblast-lineage cell is a likely target for these androgenic actions in the bone marrow[11].

Testosterone and the AR also profoundly suppress splenic B cell number in male mice and men[8]. Notably, while O-ARKO mice mimic the bone marrow B cell pattern of G-ARKO, they display unaltered numbers of mature B cells in the spleen[11]. The regulation of splenic B cell number by testosterone may therefore depend on a mechanism that acts independently of bone marrow B lymphopoiesis. One candidate mechanism may involve downregulation of the cytokine BAFF (also known as TNFSF13B), an essential survival factor for splenic B cells that is required for normal splenic B cell numbers[12]. BAFF deficiency in mice results in an arrest at the transitional B cell stage in the spleen[13] and thus a lack of mature B cells. Further, BAFF is implicated in autoimmunity, as excessive BAFF levels allow the survival of autoreactive B cells and autoantibody production[14]. Indeed, a variant in the BAFF gene has been coupled to soluble BAFF levels, blood B cell levels, and increased risk of multiple sclerosis and SLE[15]. BAFF inhibitors are approved as therapy for SLE, although their clinical usefulness remains limited[16].

In this study, we sought to define the mechanism by which testosterone regulates splenic B cell number in males. We show that testosterone is an endogenous regulator of BAFF. In line with data coupling increased splenic noradrenaline levels to depressed splenic B cell number and BAFF levels[17,18], we further show that this regulation may involve a testosterone-mediated increase in sympathetic nervous transmission[19–23]. An expansion of BAFF-producing fibroblastic reticular cells (FRCs) in spleen after castration may be coupled to reduced splenic noradrenaline levels, as an α-adrenergic agonist decreases FRC number in vitro. We conclude that the link between testosterone deficiency and increased splenic B cell numbers in males may involve nervous regulation of FRCs and BAFF.

## Results

**Testosterone regulates splenic B cell number**. First, we studied splenic B cells in mice with a general deletion of the AR (G-ARKO), where the $Ar^{flox}$ construct was recombined upon ubiquitous expression of Cre recombinase under control of the phosphoglycerate kinase-1 (Pgk1) promoter[24]. As our initial assessments showed no difference in androgen status (wet weight of androgen-sensitive organs) and splenic B cell number between $Ar^+$ and $Ar^{flox}$ male mice (Supplementary Fig. 1), $Cre^+$ littermates without the $Ar^{flox}$ construct were used as controls. Male G-ARKO mice had 1.8-fold more B cells in the spleen than littermate controls (with the Pgk1-Cre construct only) (Fig. 1a, b). By contrast, the number of dendritic cells, macrophages/monocytes and neutrophils were unaltered in G-ARKO spleens (Supplementary Fig. 2a–c).

G-ARKO mice are also testosterone-deficient[25]. To distinguish the importance of testosterone vs. AR deficiency in this model, we treated castrated control and G-ARKO mice with a physiological dose of testosterone and analysed the total number of splenic B cells. Testosterone reduced the number of splenic B cells only in the controls (Fig. 1c). Thus, this inhibitory effect is entirely AR-dependent and is not mediated by other pathways, such as conversion of testosterone to estradiol.

G-ARKO did not affect the relative number of peritoneal self-renewing B1a and B1b B cells (Fig. 1d). However, the splenic numbers of both transitional (T1, T2, T3) and mature marginal zone and follicular B cells, as well as splenic B1 cells, were increased (Fig. 1e–j). Further, G-ARKO males had a modest increase in the relative number of B cells in blood (Supplementary Fig. 2d).

In spleen sections from G-ARKO mice and controls (Fig. 1k), the total number of B cell follicles was unaltered. However, G-ARKO mice had larger follicular (Fig. 1l, m) and peri-arteriolar lymphoid sheath (PALS) areas (Fig. 1n) than controls.

We also investigated whether G-ARKO mice had increased concentrations of autoantibodies associated with autoimmune disease[26]. Indeed, while total IgG levels were slightly reduced (Fig. 1o), older G-ARKO mice had increased serum titres of IgG anti-DNA antibodies (Fig. 1p).

**Splenic mature B cell number is unaltered in O-ARKO mice**. An increased output of B cells from the bone marrow might explain the increased splenic B cell pool in androgen/AR-deficient males. We have shown that osteoblast-lineage cell-specific (O-)ARKO mice have alterations in bone marrow B cells similar to those in G-ARKO mice[11]. Thus, osteoblast-lineage cells are likely targets for androgen/AR-mediated inhibition of bone marrow B lymphopoiesis. Analysis of the splenic B cell pool of O-ARKO mice showed that the number of splenic transitional B cells was also increased in O-ARKO mice, but to a lesser extent than in G-ARKO mice (Fig. 2a, b). However, the mature B cell pool was not increased (Fig. 2c, d). Thus, the increased number of mature B cells in the spleen of G-ARKO mice is not explained by increased B lymphopoiesis and must be driven by a peripheral mechanism.

**Testosterone regulates the B cell survival factor BAFF**. Since BAFF is important for B cell survival[12], we hypothesized that testosterone negatively regulates BAFF. Indeed, serum BAFF levels were higher in G-ARKO mice than in littermate controls (Fig. 3a). Within 7 days after castration of male wildtype mice, serum BAFF was higher than in sham-castrated controls (Fig. 3b), illustrating the regulation of BAFF by endogenous testosterone. In accordance, serum BAFF was higher in female mice compared to male littermates (Fig. 3c).

Stromal cells in the spleen, in particular FRCs, are an important source of BAFF[27]. Therefore, we investigated whether testosterone regulates BAFF expression locally in the spleen. Indeed, Tnfsf13b (Baff) mRNA in the spleen was increased in G-ARKO mice (Fig. 3d). Further, a physiological testosterone

replacement reduced spleen levels of *Baff* mRNA in castrated control but not G-ARKO mice (Fig. 3e); thus, testosterone downregulates splenic *Baff* mRNA in an AR-dependent manner. Supporting a function of stromal cells, *Baff* mRNA expression was not changed in sorted splenic B cells and other splenic leucocytes after castration (Supplementary Fig. 3a, b). Immuno-histochemical analysis revealed that BAFF-positive cells were abundant and, as expected[27], mainly located in the follicles

(Fig. 3f). Consistent with the *Baff* mRNA results and the increased B cell numbers, the relative BAFF-positive area was larger in G-ARKO mice (Fig. 3g).

Since BAFF signals through the BAFF receptor (BAFF-R) for the survival of the pre-immune mature B cell pool[27], we studied the effect of castration on splenic B cell numbers during treatment with a monoclonal antibody that inhibits binding to the BAFF-R (9B9), or a control antibody (5A12)[28]. Although

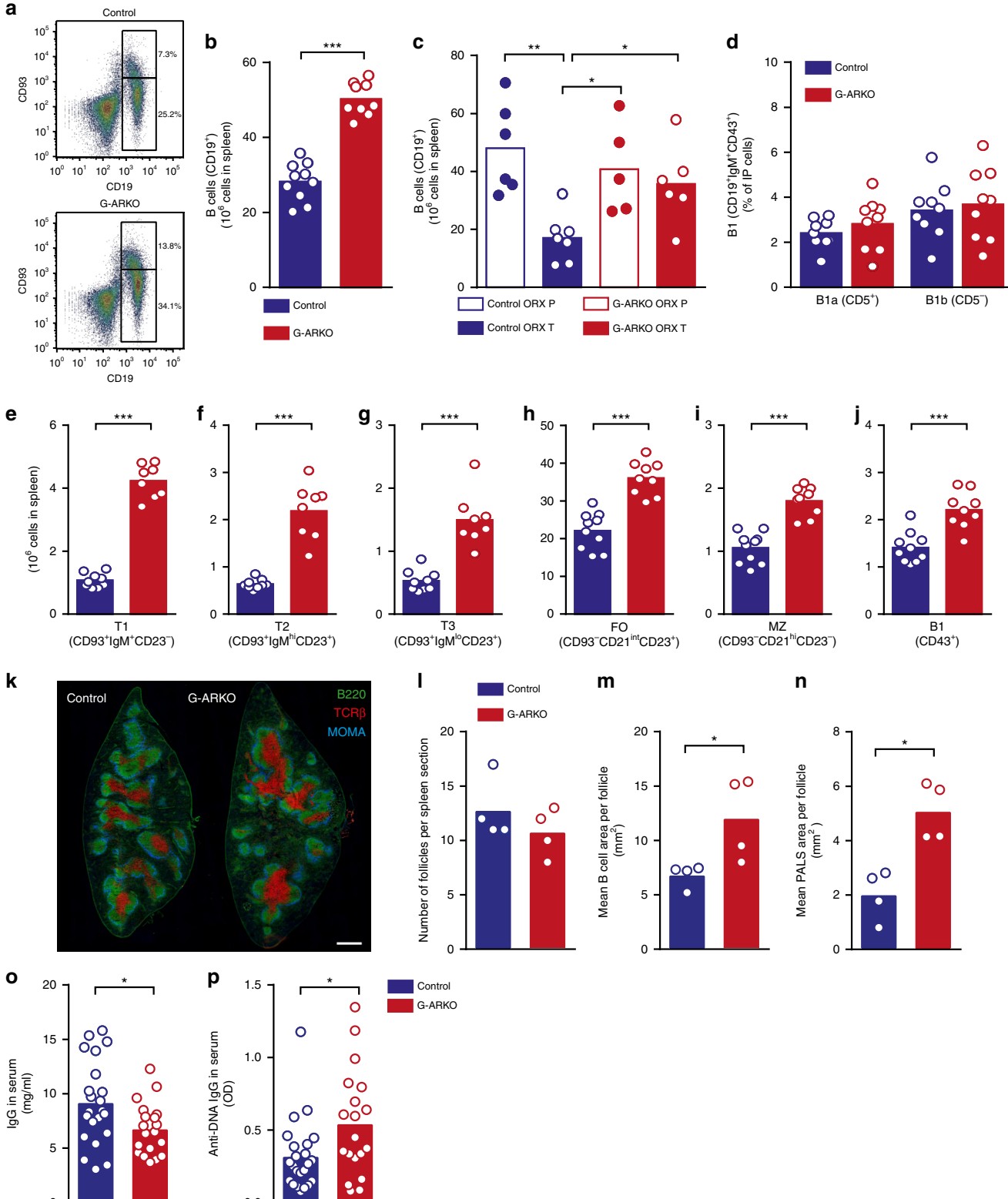

neither treatment affected transitional T1 B cells, 9B9 reduced the effect of castration on T2 and T3 B cells (Fig. 3h–j) and blocked the effect of castration on the number of mature splenic B cells (Fig. 3k). This was not explained by altered *Baffr* (*Tnfrsf13c*) mRNA expression in B cells of castrated mice (Supplementary Fig. 3c). Thus, endogenous testosterone regulates mature splenic B cell number by a BAFF-R-dependent mechanism.

To assess androgen-mediated regulation of BAFF in humans, we measured serum BAFF levels in a population-based cohort of 160 healthy men 18–50 years of age. Twelve were biochemically defined as hypogonadal and 114 as eugonadal (Supplementary Table 1); 34 did not meet the criteria for eu- or hypogonadism and were excluded. Serum BAFF levels were significantly higher in the hypogonadal men (mean ± s.d., 968 ± 285 vs. 809 ± 229 pg/ml; mean difference 160 pg/ml, 95% confidence interval 20–300 pg/ml, $P = 0.027$ by linear regression; Fig. 3l). The difference remained significant after adjustment for age and body mass index ($P = 0.011$ by linear regression). Thus, BAFF levels reflect testosterone status in men.

**AR regulation of splenic B cells is not B cell intrinsic.** The AR is expressed in B cells, and a B cell-intrinsic effect of the AR in the regulation of splenic B cell number has been proposed[29]. Therefore, we analysed splenic B cell number in two B cell-specific AR knockout (B-ARKO) mouse models generated with B cell-specific Cre constructs: *Cd79a-Cre* expressed in early pro-B cells[30] and *Cd19-Cre* expressed in pre-B cells[31]. Despite highly efficient deletion of AR in B cells, B cell numbers in the spleen were unaltered in both models (Table 1).

**Testosterone regulates splenic catecholamine levels.** We next assessed a possible connection to neuronal signalling, which profoundly affects splenic homoeostasis[32]. Reduced splenic B cell number and BAFF levels and increased splenic noradrenaline levels have been reported after thoracic spinal cord injury[17,18]. Of note, testosterone increases sympathetic nervous signalling[19–23], and noradrenaline and dopamine levels in the spleen were lower in castrated males than in sham-operated controls (Fig. 4a, b). Accordingly, immunohistochemical analysis of G-ARKO spleen sections (Fig. 4c–e) showed reduced staining of tyrosine hydroxylase-positive nerve fibres in the white pulp of G-ARKO mice (Fig. 4c).

Notably, we observed proximity between nerve fibres and smooth muscle actin (SMA)-positive[33] FRC reticulum in the central parts of the follicles (Fig. 4g), where BAFF-staining was intense (Fig. 4f). Thus, there are anatomical prerequisites for the interaction between neuron-derived adrenergic neurotransmitters and BAFF-producing stromal cells[27].

**Expansion of FRCs in testosterone or AR deficiency.** Because the central part of the follicle (i.e., the T cell zone or PALS area) is enriched in nerves, BAFF and BAFF-producing FRCs, we examined this area more closely. Although G-ARKO spleens had enlarged PALS areas (Fig. 1n), the T cell numbers were, in comparison, modestly higher (CD4 T cells: $20 ± 0.8 × 10^6$ in G-ARKO vs. $14 ± 0.5 × 10^6$ in controls, $P < 0.001$; CD8 T cells: $12 ± 0.5 × 10^6$ vs. $9.2 ± 0.4 × 10^6$, $P < 0.001$ by Mann–Whitney test). Since $CD4^+$ T cells may relay neural signals to other immune cells along the "cholinergic anti-inflammatory pathway"[34], we assessed the importance of T cells in the effect of castration on the splenic B cell population. In mice castrated before puberty, treatment with a T cell-depleting antibody regimen (anti-CD3) reduced the number of splenic $CD4^+$ T cells by 50% (Supplementary Fig. 4a); however, the effect of castration on splenic B cells was unaltered (Supplementary Fig. 4b), arguing against a central function for T cells.

We next asked whether the expansion of the PALS area was associated with expansion of the SMA-positive[33] stromal FRC compartment. Indeed, even in relation to the total PALS area, the SMA-positive area was increased in the PALS of G-ARKO mice (Fig. 5a, b). To validate the latter findings, we analysed splenic stromal cells by FACS shortly after castration (Fig. 5c). The total number of $CD45^-$ cells was unchanged in castrated mice (Fig. 5d), but the FRC population, defined as $CD45^-CD31^-$ $PDPN^+$ cells, was expanded in castrated mice (Fig. 5e, f). No similar effect was observed in inguinal lymph nodes, although these had a modest increase in B cell numbers (Supplementary Fig. 5a, b). Thus, stromal cells with BAFF-producing capacity expanded in the spleen after castration.

**Splenic FRCs express α-adrenergic receptors.** Given the proximity between nerve fibres and FRC reticulum, we next tested the hypothesis that catecholamines regulate the FRC compartment. To determine whether splenic FRCs express adrenergic receptors, we sorted splenic FRCs by FACS (Supplementary Fig. 6a, b). Our sample replicates were consistent across sorts and suggested fidelity of our sequencing data, as shown by pairs plots (Supplementary Fig. 6c). Moreover, the high levels of genes associated with fibroblastic function, including *Col1a1*, *Col1a2*, *Fbn1* and *Fn1* and low expression of haematopoietic (*Ptprc;* CD45) and endothelial (*Epcam*) genes (Supplementary Fig. 6d), showed that these cells were fibroblasts. Robust expression of classical FRC genes such as *Ccl19*, *Ccl21a*, *Cxcl13* and *Tnfsf13b* (*Baff*) suggested these were bona fide splenic FRC (Supplementary Fig. 6d). Within the adrenergic receptor family, *Adra1b*, *Adra2b* and *Adrb2* were expressed at more than one transcript per million (Fig. 6a). Other α- and β-adrenergic and dopaminergic receptors were found at lower levels and therefore were not likely expressed (Fig. 6a). To confirm α-adrenergic receptor expression at the protein level, we analysed splenic FRCs by FACS (Fig. 6b). This analysis showed that 18% (ADRA1B) and 95% (ADRA2B) of

**Fig. 1** Testosterone regulates splenic B cell number. **a** Representative plots of $CD19^+CD93^+$ transitional B cells (tB) and $CD19^+CD93^-$ mature B cells (matB) in the spleen from male control (*Pgk-Cre*$^+$) and general androgen receptor knockout (G-ARKO) mice. **b** Total $CD19^+$ B cells in the spleen in control ($n = 10$) and G-ARKO ($n = 9$) mice. **c** Total $CD19^+$ B cells in the spleen in castrated (ORX) control and G-ARKO male mice treated with placebo (P) or testosterone (T) (25 μg/day) for 4 weeks (Control ORX P, $n = 6$; Control ORX T, $n = 7$; G-ARKO ORX P, $n = 5$; G-ARKO ORX T, $n = 6$; *P*-value from Kruskal–Wallis test followed by Mann–Whitney test). **d** % B1a ($CD19^+IgM^+CD43^+CD5^+$) and B1b ($CD19^+$ $IgM^+CD43^+CD5^-$) cells in peritoneal fluid from control and G-ARKO male mice, $n = 9$/group. **e–j** $CD19^+$ B cells were divided into subpopulations; graphs show transitional T1 ($CD19^+CD93^+IgM^+CD23^-$) B cells (**e**), transitional T2 ($CD19^+CD93^+IgM^{hi}CD23^+$) B cells (**f**), transitional T3 ($CD19^+CD93^+IgM^{lo}CD23^+$) B cells (**g**), follicular (FO) ($CD19^+CD93^-CD21^{int}CD23^+$) B cells (**h**), marginal zone (MZ) ($CD19^+CD93^-CD21^{hi}CD23^-$) B cells (**i**), and B1 ($CD19^+CD43^+$) B cells (**j**) in the spleen in control ($n = 10$) and G-ARKO ($n = 9$) mice. **k** Sections of spleens from control and G-ARKO male mice. green, B cells (B220); red, T cells (TCRβ); and blue, metallophilic macrophages (MOMA/CD169). Scale bar = 500 μm. **l–n** Number of follicles per section (**l**) and mean B cell area (**m**) and mean PALS area (**n**) per follicle in spleen sections from control and G-ARKO mice, $n = 4$/group. **o** Total IgG levels in serum from control ($n = 22$) and G-ARKO ($n = 19$) mice. **p** IgG autoantibodies against DNA in control ($n = 23$) and G-ARKO ($n = 18$) mice. All bars indicate means; circles represent individual mice. *$P < 0.05$, **$P < 0.01$, ***$P < 0.001$ (Mann–Whitney test unless otherwise specified)

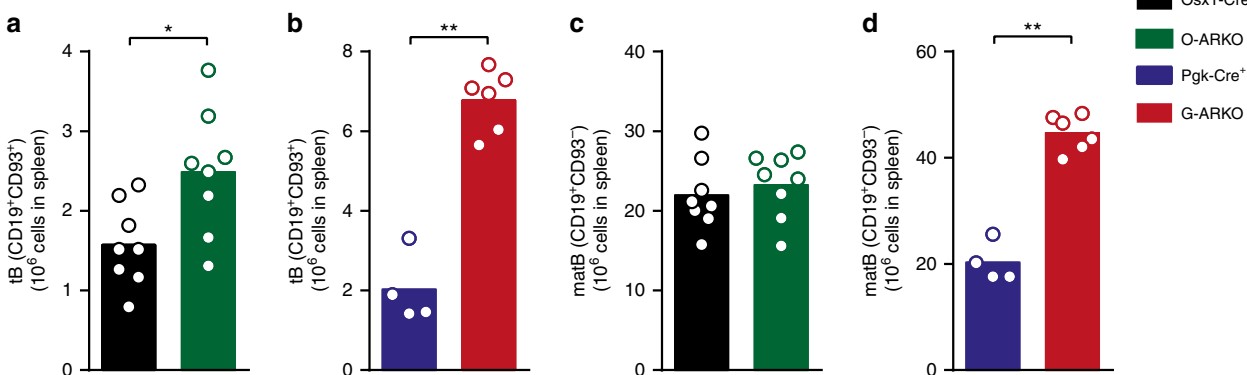

**Fig. 2** Splenic mature B cell number is unaltered in O-ARKO mice. **a** Number of CD19+CD93+ transitional B (tB) cells in spleen from control (*Osx1-Cre+*) and osteoblast lineage-specific androgen receptor knockout (O-ARKO) male mice. *n* = 8/group. **b** tB cells in spleen from control (*Pgk-Cre+*; *n* = 4) and general androgen receptor knockout (G-ARKO; *n* = 6) male mice. **c** CD19+CD93− mature B cells (matB) in spleen from control and O-ARKO mice. **d** matB cells in spleen from control and G-ARKO mice. All bars indicate means; circles represent individual mice. *$P < 0.05$, **$P < 0.01$ (Mann–Whitney test)

splenic CD45−CD31−PDPN+ cells stained positive for receptor expression.

**α-Adrenergic manipulation regulates FRC numbers in vitro.** We next set up an in vitro system to culture splenic primary stromal cells, including enrichment for CD45− and CD31− stromal cells[35] (Supplementary Fig. 7). Exposure of the enriched cells to the α-adrenergic agonist phenylephrine for 48 h reduced the number of stromal cells in a dose-dependent manner (Fig. 6c). Concomitant treatment with the α-adrenergic blocker phentolamine increased the number of cells in a dose-dependent manner (Fig. 6c). Signs of cell apoptosis were similar in the treatment groups (Fig. 6d). However, in a carboxyfluorescein succinimidyl ester (CFSE) assay, phenylephrine-treated cells had reduced proliferation, which was partly reversed by phentolamine (Fig. 6e). We confirmed that phenylephrine reduced the number of PDPN+ cells in culture and that this effect was antagonized by phentolamine (Fig. 6f, g). Thus, α-adrenergic stimulation of splenic stroma in vitro reduces the number of FRCs (CD45−CD31−PDPN+ stromal cells), and this effect is antagonized by an α-adrenergic blocker.

**Testosterone effects on splenic B cells needs neurons.** Our findings raise the question of whether interference with neurotransmission would alter the effect of castration on splenic B cell number and *Baff* mRNA. To test this possibility, we analysed the castration response in mice treated with 6-hydroxydopamine (6-OHDA)[36], which is toxic to adrenergic neurons. In saline-treated controls, the numbers of mature B cells increased in response to castration, as expected (Fig. 6h); however, 6-OHDA completely blocked this effect (Fig. 6h). Further, 6-OHDA reduced *Baff* mRNA and blocked the castration response of *Baff* mRNA in the spleen (Fig. 6i). These data show that intactness of adrenergic neurons is required for testosterone's effects on splenic B cell number and *Baff* mRNA levels.

## Discussion
Here we show that testosterone is an endogenous regulator of BAFF and propose that the link between testosterone deficiency and increased splenic B cell numbers in males involves nervous regulation of FRCs and BAFF.

Consistent with the notion that a deficiency in androgens and/or the AR increases splenic B cell number[8], both G-ARKO and castrated mice had an increased number of B cells in spleen.

Short-term castration of adult mice resulted in a doubling of the number of splenic B cells, similarly to G-ARKO mice in which the AR deletion is present from embryonic stage[24], suggesting that the regulation of splenic B cell number by testosterone/AR occurs independently of developmental effects. Our findings in O-ARKO mice did not support increased bone marrow B lymphopoiesis as a major underlying mechanism; however, we found that testosterone regulates the B cell survival factor BAFF and that BAFF-R blockade prevented the effect of castration on mature B cell number in the spleen. Moreover, serum BAFF levels were consistently increased in hypogonadal young men, a finding supported by negative correlation between serum BAFF and testosterone levels in men with psoriatic arthritis[37].

In our study, BAFF levels were modestly increased by testosterone/AR deficiency. There are no data available on the concentration-response relation between local or systemic BAFF levels and B cell numbers in mice. BAFF overexpression and knockout are both extreme models, which do not contribute to this knowledge. There are small biological variations in BAFF levels in homoeostasis, which may be concordant with a tight regulation of BAFF and that even small alterations may change B cell homoeostasis. This notion is supported by human data, suggesting that serum BAFF levels within normal ranges are inversely associated with peripheral B cell numbers[38,39]. Further, healthy subjects heterozygous for the *BAFF* variant that has been associated with clinical autoimmune disease have a modest 21–23% increase in serum BAFF levels[15].

In the present study, castration's effect on mature splenic B cell number was prevented by concomitant treatment with a blocking BAFF receptor antibody, supporting that endogenous testosterone regulates mature splenic B cell number in a BAFF-R-dependent manner. The intact effect of castration on splenic transitional T1 B cell number during BAFF-R blockade may reflect an intact, BAFF-independent[40] increase in the influx of early transitional B cells from the bone marrow to the spleen, corresponding to the increase in early transitional B cells in the spleens of O-ARKO mice[11]. Combined, these two distinctly regulated actions in bone marrow and spleen may explain why all splenic B cell subsets were increased in G-ARKO mice, but only T2 and mature B, but not T1, cells were increased in mice overexpressing BAFF[12,40].

We found here that both noradrenaline and dopamine concentrations in the spleen were lower in castrated male mice than in sham-operated controls. These findings are in line with the

reduced catecholamine levels in, for example, the hearts of androgen-deficient male mice and men[22,23] and the reduced tyrosine hydroxylase activity in sympathetic ganglia after castration[21]. Testosterone increases sympathetic nervous signalling[19–23] and likely does so at different levels: from the brain to direct modulation of autonomous reflex pathways. Studies that

attempted to decipher the effects of testosterone on pelvic autonomous pathways illustrate the complexity of this modulation[19]. These studies also show that testosterone affects noradrenaline levels not only in reproductive organs but also in other visceral organs[19], although such effects in the spleen have not been described previously.

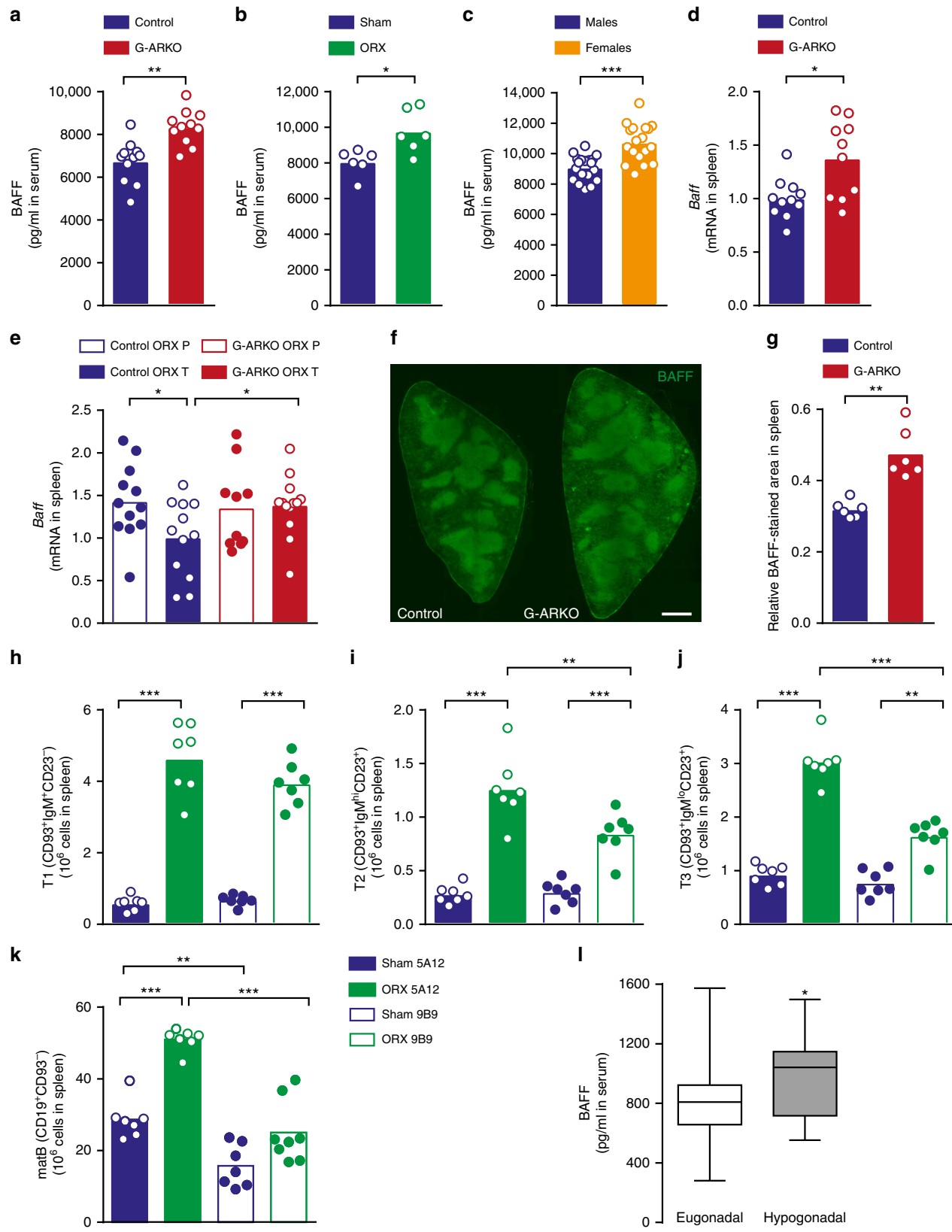

The fact that neuronal signalling has profound effects on splenic homoeostasis[32] is well accepted; for example, by the early 1980s noradrenaline from sympathetic nerve terminals was known to negatively regulate antibody responses in the spleen[41]. However, previous research mainly focused on immune cells, rather than stromal cells, as direct targets of neurotransmitters[41]. Here we found that splenic FRCs express α-adrenergic receptors and that exposure of primary splenic stromal cells to an α-adrenergic agonist reduces the expansion of these cells in vitro. Accordingly, spleens from castrated mice had low noradrenaline levels and an increased number of FRCs. To our knowledge, the regulation of FRCs or other splenic stromal cells by adrenergic transmitters has not been investigated previously. Outside the immune system, adrenergic regulation of fibroblastic cells is important in the heart[42] and lung[43], and the sympathetic nervous system regulates the proliferation of perivascular stromal cells in the bone marrow[44].

A function of the stroma in autoimmune diseases, such as Sjögren´s syndrome and rheumatoid arthritis, is increasingly recognized[45]. In primary Sjögren's syndrome, the local formation of tertiary lymphoid structures (TLS) associates with a negative prognosis. TLS contain networks of stromal cells resembling those of secondary lymphoid organs and local BAFF production is associated with resistance to B cell depletion and loss of clinical response[46]. An important question raised by our findings is whether stromal cells of TLS in Sjögren´s syndrome and other autoimmune disorders may be regulated directly by the adrenergic nervous system.

Our findings suggest that the link between testosterone/AR deficiency and increased BAFF production involves regulation of sympathetic neuronal activity by testosterone and the AR, likely at multiple levels in the autonomous nervous system, resulting in lower noradrenaline levels in the spleen. Noradrenaline mediates the reduction of FRC number through α-adrenergic receptor expression on FRCs. Thus, the decreased splenic noradrenaline levels in testosterone/AR deficiency increases the number of FRCs. Since BAFF production in secondary lymphoid organs relies heavily on the FRC stromal compartment[47], these events provide a mechanism for the increased splenic production of BAFF in testosterone/AR deficiency. The increase in B cell number in the lymph nodes was less pronounced than in the spleen, and not associated with increased FRC number, suggesting that this is a systemic effect of BAFF. Overall, our findings are consistent with recent results of depressed splenic B cell number and BAFF levels, in conjunction with increased splenic noradrenaline levels, after higher, but not lower, thoracic spinal cord injury[17,18].

While we suggest that testosterone regulates BAFF and splenic B cell number in an AR-dependent manner, the functional relevance of this regulation for antibody-mediated diseases remains unclear. G-ARKO mice displayed increased anti-DNA antibodies that may be associated with autoimmune disease[26], but we have not studied disease development in G-ARKO or castrated mice. However, castration has previously been shown to aggravate disease in different animal models of autoimmune disease, such as experimental lupus and arthritis models[4–6,48–51]. Further, treatment with both testosterone and the endogenous AR agonist dihydrotestosterone ameliorates experimental lupus and arthritis[6,48–50], supporting a protective effect of AR activation. Importantly, while BAFF inhibition may be protective in experimental lupus models[52,53], the importance of BAFF regulation for the beneficial effects of testosterone/AR agonist in experimental disease models remains unclear. Once the AR target cell for BAFF regulation is identified, a knockout of the AR specifically in this cell type may be a useful approach to address if AR signalling affects autoimmunity via BAFF regulation.

In accordance with the lower prevalence of autoimmune diseases in men, it has been proposed that estradiol accelerates and testosterone inhibits autoimmunity[1]. Interestingly, both hormones suppress bone marrow B lymphopoiesis[11,54] but differ in their effects on splenic B cell number and BAFF. In the present study, we found that serum BAFF levels were higher in female compared to male mice, which is in accordance with a previous report of sex differences in BAFF levels among wild-type controls[55]. We further found that testosterone suppresses BAFF, and others have shown that estradiol increases Baff mRNA levels and the relative number of mature B cells in the spleen[54]. Hence, estradiol elevates BAFF production and accelerates autoimmune disease[1,54], whereas testosterone has the opposite effects[8]. Like testosterone, estradiol has complex actions on autonomic neuroeffector mechanisms, but its net effect is likely reduced peripheral sympathetic nerve activity[56]. Therefore, differences in the regulation of BAFF by estradiol and testosterone, potentially driven by nervous mechanisms, may contribute to the sex difference in autoimmune diseases in which BAFF has a pathogenic function.

In conclusion, our study uncovers a previously unrecognized regulation of BAFF by testosterone. Our data raise important questions about BAFF in testosterone-mediated protection against autoimmunity as well as the sexual dimorphism in autoimmune disorders.

## Methods

**Animals and study design.** Male mice with a general depletion of the AR from the embryonic stage (G-ARKO)[24] were generated by breeding $Ar^{+/flox}$ females[24] with males expressing Cre recombinase ubiquitously under the control of the $Pgk1$ promoter[24,25]. Osteoblast-lineage cell-specific (O-ARKO)[11] and B cell-specific (B-ARKO) ARKO mice were generated by breeding $Ar^{+/flox}$ females with $Osx1$-$Cre^+$ males (Tg(Sp7-tTA,tetO-EGFP/cre)1Amc, Jackson Laboratory) to produce O-ARKO, $Cd79a$-$Cre^+$ mice (also called $Mb1$-$Cre^+$; from Prof. Michael Reth)[30] or with $Cd19$-$Cre^+$ mice (B6.129P2(C)-$Cd19^{tm1(cre)Cgn}$/J, Jackson Laboratories) to produce B-ARKO mice. $Ar$, $Cre$, and $Zfy$ (for gender) expression was assessed by PCR amplification of genomic DNA (primer sequences in Supplementary Table 2). In all experiments, mice were compared to littermate controls; all mice were on

**Fig. 3** Testosterone regulates the B cell survival factor BAFF. **a** Serum BAFF level in control ($Pgk$-$Cre^+$) and general androgen receptor knockout (G-ARKO) male mice, $n = 10$/group. **b** Serum BAFF level in castrated (ORX) and sham-operated male mice 7 days after surgery, $n = 6$/group. **c** Serum levels of BAFF in serum from male ($n = 19$) and female ($n = 18$) 12-week-old littermate wild-type mice. **d** Baff mRNA expression in the spleen in control and G-ARKO male mice, $n = 10$/group. **e** Baff mRNA expression in the spleen from castrated (ORX) control and G-ARKO male mice treated with placebo (P) or testosterone (T) (25 μg/day) for 4 weeks (Control ORX P, $n = 12$; Control ORX T, $n = 12$; G-ARKO ORX P, $n = 10$; G-ARKO ORX T, $n = 13$; P-value from Kruskal–Wallis test followed by Mann–Whitney test). **f** Sections of spleens from control and G-ARKO male mice. Green, BAFF staining. Scale bar, 500 μm. **g** Quantification of BAFF-stained area/total area of spleen section, $n = 6$/group. **h–k** Transitional T1 (CD19$^+$CD93$^+$IgM$^+$CD23$^-$) B cells (**h**), transitional T2 (CD19$^+$CD93$^+$IgM$^{hi}$CD23$^+$) B cells (**i**), transitional T3 (CD19$^+$CD93$^+$IgM$^{lo}$CD23$^+$) B cells (**j**) and mature B cells (matB) (CD19$^+$CD93$^-$) (**k**) in spleens of castrated (ORX) or sham-operated male mice treated with a BAFF-R blocking (9B9) or control (5A12) antibody (Sham 5A12, $n = 7$; ORX 5A12, $n = 7$; Sham 9B9, $n = 7$; ORX 9B9, $n = 8$; P-value from Kruskal–Wallis test followed by Mann–Whitney test). **a–k** Bars indicate means; circles represent individual mice. *$P < 0.05$, **$P < 0.01$ and ***$P < 0.001$ (Mann–Whitney test unless otherwise specified). **l** Serum BAFF levels in healthy young men biochemically defined as eugonadal (total testosterone > 13 nmol/l and free testosterone > 0.22 nmol/l; $n = 114$) or hypogonadal (total testosterone < 13 nmol/l and free testosterone < 0.22 nmol/l; $n = 12$). Data are presented as a box plot (25th–75th percentile). Horizontal bars represent median values and whiskers represent minimum and maximum values. *$P < 0.05$ (linear regression)

**Table 1 AR regulation of splenic B cells is not B cell-intrinsic**

| Target cell (Cre promoter) | Mean reduction of *Ar* exon 2 DNA in B cells | *P*-value | No. of B cells (×10⁶) | | *P*-value |
| --- | --- | --- | --- | --- | --- |
| | | | **Controls** | **B-ARKO mice** | |
| B cells (*Cd79a*) | 93% | <0.0001 | 24 ± 4 | 20 ± 2 | 0.35 |
| B cells (*Cd19*) | 86% | <0.0001 | 20 ± 3 | 20 ± 2 | 0.88 |

Presentation of B cell-specific ARKO model, Cre promoter used, AR knockout level in the model (measured as relative level of *Ar* exon 2 DNA in the B cell compartment), and total number of CD19⁺ B cells in spleen in male control and B cell-specific ARKO (B-ARKO) mice. $n = 11$ and 8 (*Cd79a-Cre*⁺ controls and corresponding B-ARKO), $n = 10$ and 11 (*Cd19-Cre*⁺ controls and corresponding B-ARKO). Values represent mean ± s.e.m. *P*-values from Mann–Whitney test.

C57BL/6J background, unless otherwise stated. The mice were housed in specific pathogen-free facility, in a temperature- and humidity-controlled room with a 06:00–18:00 h light cycle and fed a soy-free diet (R70, Lantmännen) and tap water ad libitum. All animal studies were approved by the Ethics Committee on Animal Care and Use in Gothenburg or at Genentech and were in accordance with the NIH "Guide for the Care and Use of Laboratory Animals" (8th ed. The National Academies Press. 2011).

**Orchiectomy and testosterone treatment**. Four weeks before tissue collection, 8-week-old G-ARKO and littermate controls were bilaterally orchiectomized or sham-operated and implanted subcutaneously with a small slow-release pellet containing placebo or a physiological dose[25] of testosterone (25 μg/day; Innovative Research of America). In separate experiments, male wild-type mice were sham-operated or orchiectomized 7–14 days before tissue collection.

**Treatment with BAFF-R blocking antibody**. Twelve-week-old male wild-type mice were bilaterally orchiectomized or sham-operated and given a single intravenous injection (0.5 mg) of a monoclonal BAFF-R blocking antibody (9B9) or a control non-blocking antibody (5A12), both provided by Prof. Rolink[28]. Spleens were collected 2 weeks later. Previous studies have shown that B cell reduction by 9B9 is neither due to antibody-dependent cellular cytotoxicity nor to complement-mediated lysis[28].

**T cell depletion**. At 4 weeks of age, male mice (B6.129P2-*Apoe*[tm1Unc;] Taconic) were bilaterally orchiectomized or sham-operated. One week later, the mice received an intraperitoneal injection (50 μg) of monoclonal anti-mouse CD3 antibody (145-2C11 f(ab′)2 Fragments; BioXCell) or a control antibody (Hamster IgG f(ab′)2 Fragments; BioXCell) on 5 consecutive days. The injections were repeated at 3-week intervals (when the mice were 5, 8, 11 and 14 weeks old). Spleens were collected at 16 weeks of age.

**Treatment with 6-hydroxydopamine**. Eight-week-old male wild-type mice were bilaterally orchiectomized or sham-operated and treated with a single intraperitoneal injection of 6-hydroxydopamine (250 mg/kg; Sigma Aldrich) or saline[36]. Tissues were collected 2 weeks later.

**Tissue collection and cell isolation**. All mice were killed at 10–14 weeks of age (unless otherwise specified). The mice were anaesthetized (Isoflo vet; Orion Pharma Animal Health), blood was drawn from the left ventricle and the mice were perfused with saline at physiological pressure (around 110 mmHg) for 5 min. Cells from the peritoneal cavity were isolated by lavage and kept in phosphate-buffered saline (PBS) on ice. After dissection, tissues were snap-frozen in liquid nitrogen for DNA and RNA quantification, frozen for cryosectioning (Cryomount, Histolab Products) or kept in PBS on ice for cell isolation.

**Preparation of cells for flow cytometry**. To obtain single cells, spleen and inguinal lymph nodes were directly passed through a nylon wool sieve (70 μm). To obtain stromal cells, the spleen and inguinal lymph nodes were minced and enzymatically digested, as described[35]. Erythrocytes (in spleen) were haemolysed (0.16 M NH₄Cl, 0.13 M EDTA and 12 mM NaHCO₃ in H₂O or ACK lysis buffer), and the cells were washed in flow cytometry buffer (2% foetal bovine serum and 2 mM EDTA in PBS), filtered through a 70 μm cell strainer and counted in an automated cell counter (Sysmex or ViCell). After Fc-blockage (anti-mouse CD16/CD32, clone 2.4G2, BD Biosciences), expression of various cell-surface markers was detected with fluorochrome-conjugated antibodies: CD19 (1D3; BD Biosciences), CD93 (AA4.1; eBioscience), IgM (Polyclonal; SouthernBiotech), CD23 (B3B4; BD Biosciences), CD21 (7G6; BD Biosciences), CD43 (S7; BD Biosciences), CD11c (N418; Biolegend), CD11b (M1/70; BD Biosciences), F4/80 (BM8; Biolegend), Gr1 (RB6-8C5; eBioscience), CD45 (30-F11; BioLegend), CD31 (390; BioLegend), podoplanin (8.1.1; BioLegend), Pdgfrα (AP5; Biolegend), rabbit anti-α1b adrenergic receptor (EPR10336; Abcam), rabbit anti-α2b adrenergic receptor (EPR9623; Abcam), rabbit IgG monoclonal isotype control (EPR25A; Abcam), F(ab′)₂ fragment donkey anti-rabbit IgG (H + L) Alexa Fluor 647 (711-606-152; Jackson ImmunoResearch). For exclusion of nonviable cells after enzymatic

digestion, LIVE/DEAD Fixable Aqua Dead Cell stain (Molecular Probes) was added.

Peritoneal cells were isolated by lavage and washed with PBS. Expression of cell-surface markers was detected with fluorochrome-conjugated antibodies after Fc-blockage (anti-mouse CD16/CD32; BD Biosciences): CD19 (1D3; BD Biosciences), IgM (Polyclonal; SouthernBiotech), CD43 (S7; BD Biosciences), and CD5 (53-7.3; BD Biosciences).

Blood was drawn from the left ventricle and prevented from clotting by storage in EDTA-coated microvette tubes (20.1288; Sarstedt). Leucocytes were enriched by haemolysis of erythrocytes and the cells were washed in flow cytometry buffer. After Fc-blockage, expression of CD19 cell-surface marker was detected with a fluorochrome-conjugated antibody, as described above.

Fluorochrome-minus-one was used as control in all flow cytometry experiments. Cells were analysed with a FACS Canto II, LSRII, FACS Aria, or Accuri C6 (all BD Biosciences); the data were further analysed with FlowJo software (Tree Star). Cells were sorted directly into Trizol (Thermofisher) after a purity check on a FACS Aria or FACS Fusion (BD Biosciences).

The gating strategies for splenic B cells (Supplementary Fig. 8a), peritoneal B cells (Supplementary Fig. 8b), and splenic FRCs (Supplementary Fig. 8c) are described in the Online Supplement.

**Sorting of splenic leucocytes**. From isolated splenic cells, B cells were sorted using anti-mouse CD19-conjugated magnetic microbeads (#130-052-201; Miltenyi Biotec), according to the manufacturer's instructions. From the CD19⁻ fraction CD45⁺ leucocytes were isolated using anti-mouse CD45-conjugated magnetic microbeads (#130-052-301; Miltenyi Biotec).

**Antibodies in serum**. Serum antibody levels were measured in 34-week-old G-ARKO and control mice by enzyme-linked immunosorbent assay (ELISA)[57]. For total IgG, ELISA plates were coated with anti-Ig (H + L)(ASB-103101, Nordic Biosite). Purified mouse IgG (0107-01, SouthernBiotech) was used as standard to allow exact calculation of IgG levels. For anti-DNA IgG, ELISA plates were pre-coated with methylated bovine serum albumin and calf thymus DNA (Sigma-Aldrich). The antibody reactivity in serum was measured with an alkaline phosphatase-conjugated anti-mouse IgG antibody (1030-04; SouthernBiotech). The serum was serially diluted; for total IgG we present data at dilution 1:250,000 and for anti-DNA IgG at 1:50. All samples were run in duplicate and corrected for background binding.

**Spleen immunohistochemistry**. Cryosections (10 μm) of spleen were air dried for 2 h and stored at −20 °C. For immunolabelling, sections were fixed for 5 min with 2% formaldehyde, permeabilized with 0.1% Triton-X for 4 min, treated with Avidin/Biotin Blocking Kit (Cat# BC-AB972; BioCare Medical), and incubated with 1% bovine serum albumin in PBS for 15 min before addition of primary antibodies. PBS was used for all washes throughout the immunolabelling procedures except for the last wash before mounting, which was done in deionized water. Spleen tissue sections were incubated with rat anti-mouse CD169 (3D6.112; AbD Serotec; 1:100) overnight at 4 °C and then with F(ab)2 AF594-conjugated donkey anti-rat IgG (Jackson ImmunoResearch; 1:400) for 45 min at 20 °C. To block unspecific binding of the anti-rat secondary antibodies, the sections were incubated for 15 min with nonspecific rat IgG2a (R35-95; BD Biosciences; 1:30). The sections were then incubated with AF488-conjugated rat anti-human/mouse B220 (RA3-6B2; Affymetrix; 1:100) and biotinylated hamster anti-mouse TCRβ (H57-597; Affymetrix; 1:100) for 60 min at 20 °C and with Atto425-conjugated streptavidin (ATTO-TEC; 1:200) for 30 min and mounted with ProlongGold mounting medium (Life Technologies). In a similar protocol, sections were blocked in 1% bovine serum albumin together with Fc-receptor blockage (anti-mouse CD16/32, 2.4G2; BD Biosciences; 1:100) and incubated overnight with rat anti-mouse BAFF-AF488 (121808; R&D Systems; 1:20), rat anti-mouse IgD-biotin (11-26; SouthernBiotech; 1:150), mouse anti-smooth muscle alpha actin-Cy3 (1A4; Sigma Aldrich; 1:4000), and rabbit anti-mouse tyrosine hydroxylase (cat no. ab112; Abcam; 1:300). Sections were then incubated for 1 h with donkey anti-rabbit-AF647 (Jackson Immunoresearch; 1:300) and streptavidin-BV480 (BD Biosciences; 1:200). The number of follicles per section and areas of B cell staining were quantified by computerized

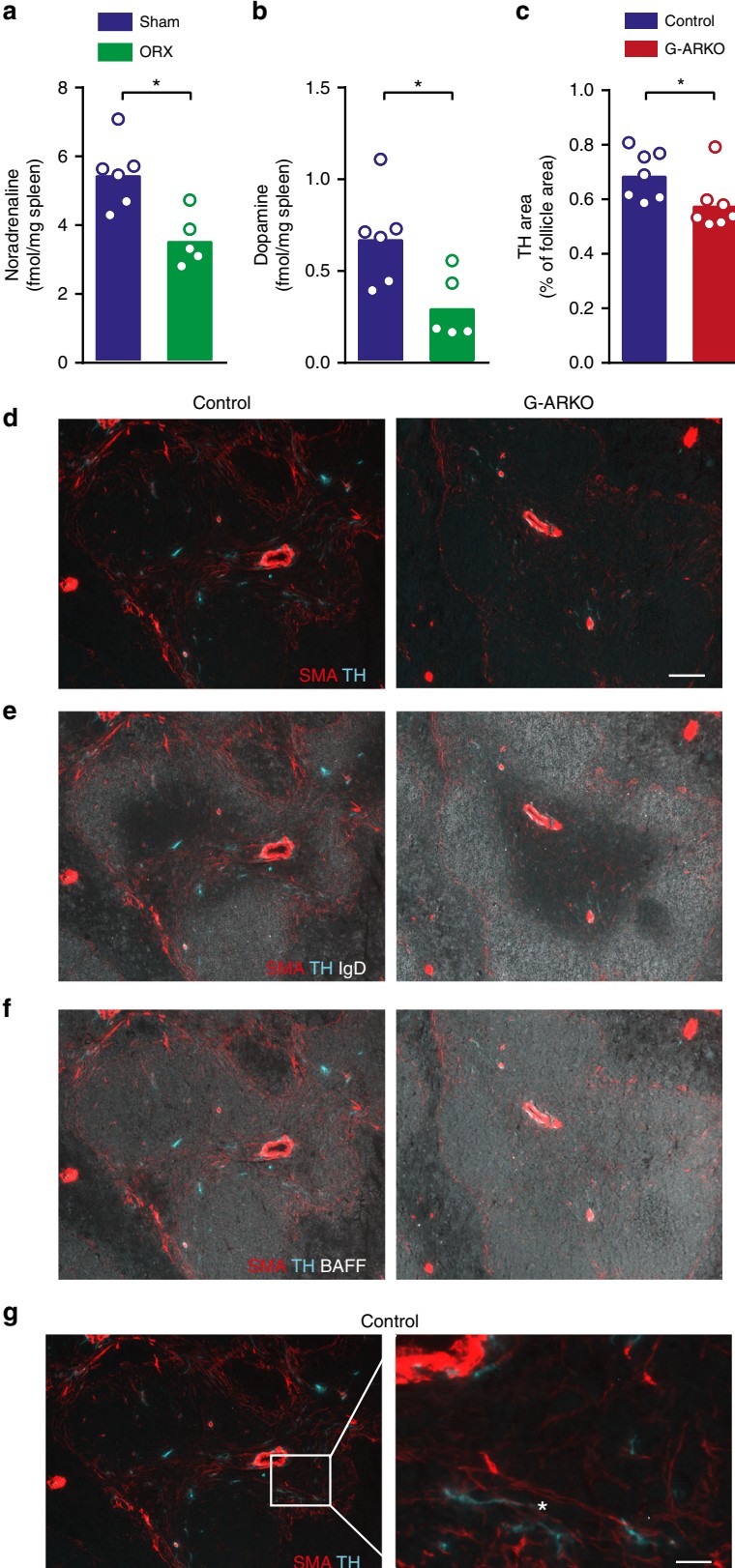

**Fig. 4** Testosterone regulates splenic catecholamine levels. **a** Noradrenaline and **b** dopamine concentration in spleens of castrated (ORX; $n = 6$) and sham-operated ($n = 5$) male mice. **c** Quantification of tyrosine hydroxylase (TH)-stained area in control (*Pgk-Cre*[+]) and general androgen receptor knockout (G-ARKO) male mice, expressed as percentage of follicle area. $n = 7$/group. **d–f** Sections of spleens from control and G-ARKO mice. Turquoise, nerve structures (tyrosine hydroxylase; TH); red, fibroblastic reticular cells (FRCs) and vascular structures (smooth muscle a-actin; SMA). B cells (IgD) are stained white in **e** and BAFF is stained white in **f**. Scale bar, 100 μm. **g** Higher-magnification image of SMA-positive FRC (red) and TH-stained nerve structures (turquoise); asterisk indicates area of co-localization. Scale bar, 20 μm. Bars indicate means; circles represent individual mice. *$P < 0.05$ (Mann–Whitney test)

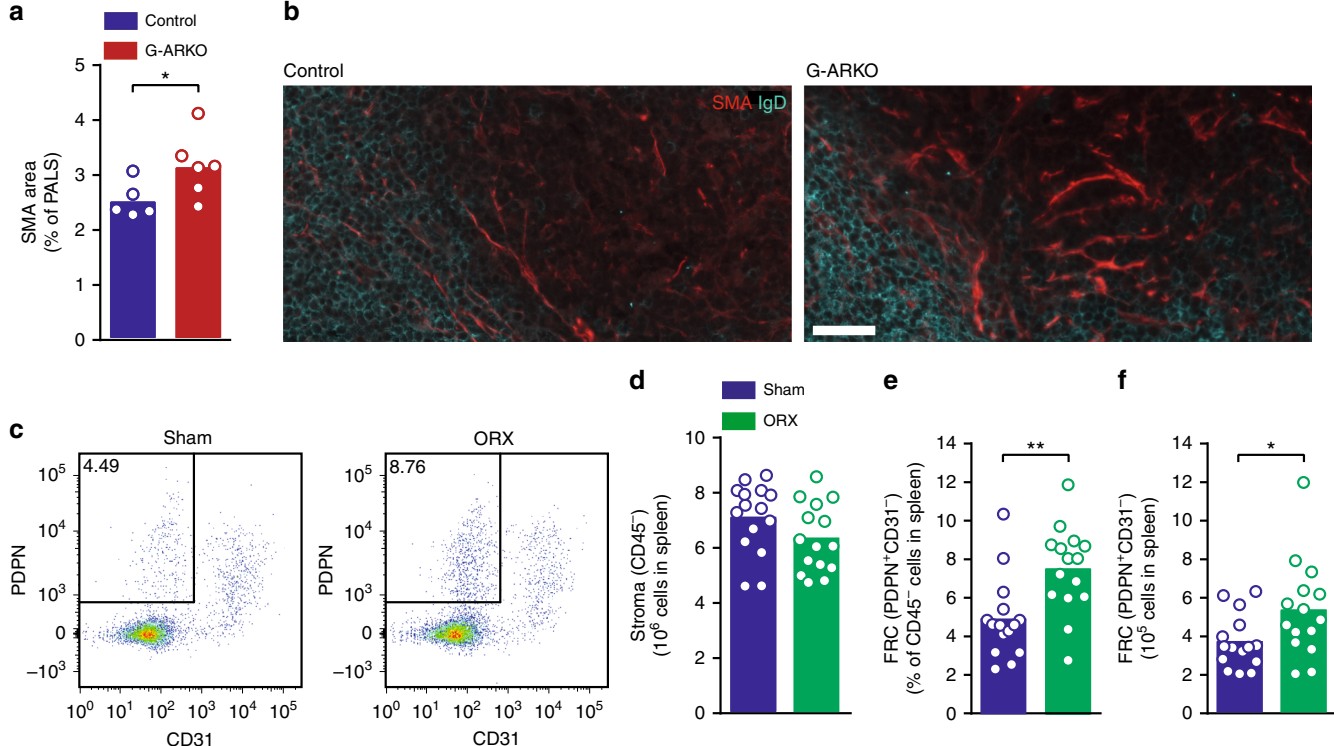

**Fig. 5** Expansion of FRCs in testosterone/AR deficiency. **a** Quantification of smooth muscle a-actin (SMA)-stained area, expressed as percentage of peri-arteriolar lymphoid sheath (PALS) area in control (*Pgk-Cre*⁺) and general androgen receptor knockout (G-ARKO) male mice. *n* = 5–6/group. **b** Section of spleens from control and G-ARKO male mice, at the border of the B cell zone and PALS area. Red, SMA-positive fibroblastic reticular cells (FRC); turquoise, IgD-positive B cells. Scale bar, 30 µm. **c** Representative plots of PDPN⁺CD31⁻CD45⁻ FRCs in the spleen from sham-operated and castrated (ORX) male mice 2 weeks after surgery. **d** Total number of CD45⁻ cells in spleens of castrated or sham-operated male mice. *n* = 15/group. **e** Frequency of FRC among stromal (CD45⁻) cells and **f** total FRCs per spleen in ORX and sham-operated male mice. Bars indicate means; circles represent individual mice.*P < 0.05, **P < 0.01 (Mann–Whitney test)

colour selection (BioPix iQ 2.2.1); the BAFF immune-labelled area was quantified by Visiopharm image analysis (version 4.6.3.857; Visiopharm).

**DNA quantification.** Since exon 2 of the *Ar* gene is excised in the ARKO mouse model[24], we quantified exon 2 by using exon 3 as a reference to assess the efficacy of the cell-specific knockouts. For this purpose, CD19⁺ cells were isolated by positive selection with Macs CD19 microbeads (Miltenyi Biotec). Genomic DNA from CD19⁺ cells, and organs/tissues was isolated with the DNeasy blood and tissue kit (Cat#69504; Qiagen) according to the manufacturer's instructions. Amplification of genomic DNA was detected with SyBR green master mix (Cat#4367659; Thermofisher Scientific) and an ABI Prism 7900HT Sequence Detection System (Applied Biosystems). The following primer pairs were used: *Ar* exon 2: forward GGACCATGTTTTACCCATCG and reverse CCACAAGTGA-GAGCTCCGTA; and *Ar* exon 3: forward TCTATGTGCCAGCAGAAACG and reverse CCCAGAGTCATCCCTGCTT. Ct values for *Ar* exon 2 were normalized to Ct values for *Ar* exon 3 by the $2^{-\Delta\Delta ct}$ method.

**Serum BAFF in mice and healthy men.** Circulating BAFF concentrations in mouse serum (diluted 1:6) were determined with a Quantikine ELISA (MBLYS0, R&D Systems) according to the manufacturer's instructions.

In an observational study, fasting serum samples were obtained before 10 a.m. from the control cohort of a study on subfertility in men[58]. This group of control men (18–50 years old at inclusion) was selected from the Swedish Population Register. None of the participants reported any serious medical problems, and none were receiving fertility treatment, anti-diabetic/insulin therapy, opioids, or glucocorticoids or other anti-inflammatory drugs. The study was approved by the Regional Ethical Review Board at Lund University, Sweden. All subjects gave written informed consent. Serum for assessment of serum testosterone and BAFF was available for 160 men. Serum testosterone and free testosterone were assessed as described[58]. BAFF concentrations in human serum (diluted 1:2) were determined with a Quantikine ELISA (DBLYS0B R&D Systems). Hypogonadism was biochemically defined as total testosterone <13.0 nmol/l plus free testosterone <0.22 nmol/l[59]. Eugonadism was defined as total testosterone >13.0 nmol/l plus free testosterone >0.22 nmol/l.

**RNA isolation and real-time RT-PCR.** Total RNA was extracted from spleen or isolated spleen cells with the RNeasy Mini Kit (Cat#74104; Qiagen) according to the manufacturer's instructions. To eliminate DNA contamination, a DNase I (Qiagen) step was included. Complementary DNA was synthesized from total RNA with a high-capacity cDNA reverse transcription kit (Cat#4374966; Applied Biosystems). RT-PCR analysis was done with predesigned TaqMan Gene Expression Assays (Applied Biosystems): *Tnfsf13b* (*Baff*; Mm00446347_m1) or *Tnfrsf13c* (*Baffr*; Mm00840578_g1) (FAM) with *Hprt* (Mm00446968_m1, VIC) as a reference gene. The analyses were run in an ABI Prism 7900HT Sequence Detection System or a ViiA 7 (Applied Biosystems). Data were normalized to the reference gene, and gene expression was calculated with the $2^{-\Delta\Delta ct}$ method.

**Measurement of splenic catecholamines.** Noradrenaline and dopamine levels in the spleen were determined by high-performance liquid chromatography followed by electrochemical detection. Spleens in a solution of 0.1 M perchloric acid, 5.37 mM EDTA, and 0.65 mM glutathione were homogenized with a Branson Sonifier 450 (Branson Sonic Power). After centrifugation (10,000 × *g*, 5 °C, 10 min), the supernatant was collected and immediately analysed for dopamine and noradrenaline. The amines were chromatographed on an ion-exchange column (EC 150/2 Nucleosil 100-5 SA, Macherey-Nagel) with a mobile phase consisting of 13.3 g of citric acid, 5.84 g of NaOH, 40 mg of EDTA, and 200 ml of methanol in distilled water to a total volume of 1000 ml. Dopamine and noradrenaline were detected at a 0.55 V oxidizing potential with a Waters 460 detector (Millipore Waters, Milford). The resulting currents were acquired and integrated with Chromeleon software (Thermofisher Scientific).

**RNAseq and bioinformatic analysis.** Male C57BL/6 mice 7–9 weeks old (Jackson Laboratories) were used for RNAseq experiments. For splenic FRC ex vivo quantification, 6–8-week-old male wild type or Ccl19-cre^neg×Rosa26-DTR^+/− mice were used. Each splenic FRC sample was derived by pooling lysed cells from three independent experiments in which 2500 to 12,000 cells were sorted. In all cases, samples were sorted at >94% purity. RNA was isolated as described[60]. All samples had RIN scores of 8.7 or greater, as determined with a BioAnalyzer (Agilent). Paired-end RNA-seq libraries were constructed from 747 pg of RNA using the

SMARTer + NexteraXT Low Input RNA Kit. Libraries were then sequenced on an Illumina HiSeq yielding, on average, 35 million read pairs ($2 \times 50$ bp) per sample. We used Salmon version 0.6 (ref. [61]) to quantitate reads against the GENCODE M9 mouse transcript models with "--biasCorrect" enabled and the "-l IU" library type. These data were imported into R using the tximport package[62] to get gene level expression estimates, and limma/voom[63] was used for differential expression analysis.

**Splenic FRC culture in vitro**. Spleens from male mice (B6SLJ; Taconic) were dissected, minced, and enzymatically digested[35]. The cells were filtered through a nylon wool sieve (70 μm) and incubated overnight in α-minimal essential medium (32571-028; Thermofisher Scientific), supplemented with 10% foetal bovine serum and 1% penicillin/streptomycin. Non-adherent cells were removed by washing with PBS, and the attached cells were cultured for 6 days. The cells were enriched for non-haematopoietic and non-endothelial cells by anti-CD31 and -CD45

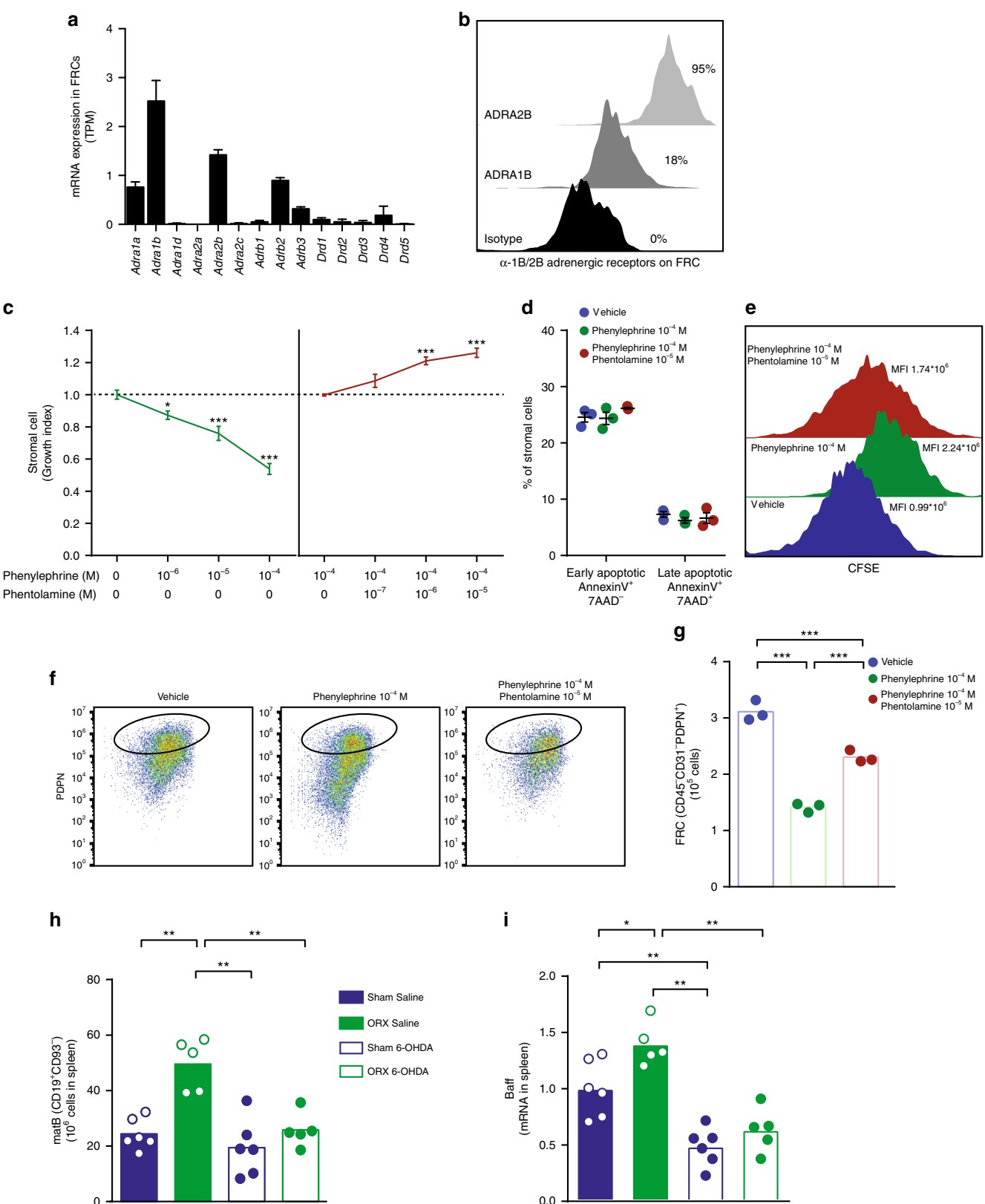

microbeads (#130-097-408 and #130-052-301; Miltenyi Biotec) according to the manufacturer's instructions. The enriched stroma was counted and plated overnight in six-well plates at a density of 0.2 million cells per well. Before and after enrichment, the stromal cell composition was assessed by flow cytometry for CD31 (390; eBioscience), CD45.1 (A20; BD Biosciences), and podoplanin (8.1.1; BioLegend). Fluorochrome-minus-one was used as control, and dead cells excluded from analyses by positivity for 7AAD (A1310; Life Technologies). The cells were analysed on an Accuri C6 flow cytometer (BD Biosciences), and the data were analysed with FlowJo software (Tree Star).

To evaluate the effect of α-adrenergic signalling on stromal cell growth, medium containing the α-adrenergic agonist phenylephrine hydrochloride ($10^{-7}$–$10^{-4}$ M; P1250000; Sigma Aldrich) and/or the α-adrenergic antagonist phentolamine hydrochloride ($10^{-7}$–$10^{-5}$ M; P7547; Sigma Aldrich) was added to the culture and exchanged after 24 h. After 48 h, live stromal cells were counted with DAPI-containing solution #13 in a Nucleocounter NC-3000 (Chemometec). Cell viability was not affected by the α-adrenergic agonist/antagonist at the concentrations shown; however, phentolamine at $10^{-4}$ M was toxic, resulting in no live cells at 48 h. The cells were stained with anti-podoplanin antibody and analysed by flow cytometry as described above. To assess apoptosis, cells were stained with annexin V (FITC Annexin V Apoptosis Detection Kit, 556547; BD Biosciences) after a 48-h incubation with agonist/antagonist. Cell division was revealed by CFSE staining; cells were stained with CFSE (CellTrace CFSE Cell Proliferation Kit, C34554; Thermofisher Scientific) according to the manufacturer's instructions, incubated with agonist/antagonist, and analysed by flow cytometry after 48 h.

**Statistical analysis.** For statistical evaluations, Prism (version 5; GraphPad Software) and SPSS (version 15.0; SPSS) were used. For mouse data, the non-parametrical Mann–Whitney test was used for two-group comparisons, and the Kruskal–Wallis test followed by the Mann–Whitney test was used for four-group comparisons. In vitro data were analysed by one-way ANOVA followed by Bonferroni correction. Linear regression models were used to assess differences in serum BAFF levels between hypogonadal and eugonadal men with and without adjustment for age and body mass index (log-transformed). No statistical methods were used to predetermine sample size. Animals were allocated to experimental groups by genotype or arbitrarily without formal randomization and there were no exclusions of mice. Investigators were not formally blinded to group allocation during the experiment. $P < 0.05$ (two-sided) was considered statistically significant.

**Data availability.** The datasets analysed during the current study are available from the corresponding author on reasonable request. BioSample metadata are available in the NCBI BioSample database under accession numbers SAMN08824665 [https://www.ncbi.nlm.nih.gov/biosample/?term=SAMN08824665] and SAMN08824666 [https://www.ncbi.nlm.nih.gov/biosample/?term=SAMN08824666].

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

## Acknowledgements

The authors thank Annelie Carlsson, Christina Ullström and Annika Lundqvist for excellent research assistance; Erik Larsson for help with *Ar* genomic DNA primer design; Michael Reth for providing the *Cd79a-Cre*[+] mice; Aleksander Giwercman for serum from the male cohort, and Lillemor Mattsson Hultén for advice regarding ELISA analyses.

This study was supported by the Swedish Research Council, the Swedish Heart-Lung Foundation, Avtal om Läkarutbildning och Forskning (ALF) research grant in Gothenburg, the Marianne and Marcus Wallenberg Foundation, AFA Insurance, the Novo Nordisk Foundation, the Swedish Rheumatic Foundation, the King Gustav V 80-year Foundation, the Inga-Britt and Arne Lundberg Foundation, and the Torsten Söderberg Foundation. Work in the Porse lab was supported by a Centre grant from the NovoNordisk Foundation (The NovoNordisk Foundation Section for Stem Cell Biology, DanStem; Grant Number NNF15CC0027852).

## Author contributions

A.S.W., M.L.R., A.S., P.F., I.J., M.B.B., S.L., V.N.K., M.E.J., J.B.F., A.D., P.T., A.C., B.T.P., A.G.R., H.N., S.J.T., H.C., I.L.M., M.C.K. and Å.T. designed the studies or analyses. A.S.W., M.L.R., A.S., P.F., I.J., M.B.B., J.B.F., A.D., P.T. and H.N. conducted experiments and/or acquired data. A.S.W., M.L.R., A.S., P.F., I.J., M.B.B., S.L., M.E.J., A.D., P.T., H.N., I.L.M. and Å.T. analysed the data. A.S.W., M.C.K. and Å.T. wrote the manuscript. A.S.W., M.L.R., A.S., P.F., I.J., M.B.B., S.L, V.N.K., M.E.J., J.B.F., A.D., P.T., A.C., B.T.P., A.G.R., H.N., S.J.T., H.C., I.L.M., M.C.K. and Å.T. revised the manuscript.

## Additional information

**Competing interests:** The authors declare no competing interests.

