## [Peer Review File · Nature Communications]

Reviewers' comments:

Reviewer #1 (SLE, mouse models, BAFF)(Remarks to the Author):

This manuscript has already been reviewed by two expert reviewers and shows an interesting relationship between serum testosterone and BAFF levels that appears to be regulated by the neural reflex. While the physiologic significance of this finding is not yet known, the data shown in the manuscript is clear and the responses to the reviewers are detailed and in accordance with what is known about the physiology of BAFF with respect to B cells. The authors have clearly defined the scope of this manuscript and have appropriately answered those points within the paper's scope.

Reviewer #2 (Hormone/immune crosstalk)(Remarks to the Author):

The primary goal of this manuscript was to determine the direct effects of androgen receptor signaling on B cell activity, which could have implications for better understanding protection of males from autoimmune diseases, including rheumatoid arthritis. The authors show that the impact of androgen receptor signaling on B cells is mediated by B-cell activation factor (BAFF), which is a survival factor for B cells. Males have lower levels of BAFF than females and castration of males increases concentrations. Deletion of androgen receptors completely or specifically in osteoblast resulted in elevated levels of BAFF illustrating that androgen receptor signaling regulates BAFF, which in the spleen is associated with regulation of noradrenaline.

Major

1. Functional significance. From the previous review, the reviewers requested that the authors downplay the translational aspects of these studies. The fundamental problem with this paper is not that they need to downplay the functional significance, but rather that they have to show the functional significance for disease. They attempt this by measuring anti-bovine DNA responses and show that these antibodies are elevated in complete AR knock out mice. But then, in Suppl. Fig 2, they show that these animals have more B cells, but lower IgG antibody titers. To make their observed effects of AR signaling on BAFF significant, they would need to show that this is relevant to an antibody-mediated disease, such as rheumatoid arthritis, for which there are animal models. Without this, this observation lacks significance and with discrepant findings buried in the supplemental figures (being added from the previous review) it is difficult to determine if any of the reported effects would matter for a disease state. In its current state, the entire paper is a study of how AR signaling affects baseline B cells, which to be honest is often quite marginal. There may be larger effects if B cells were challenged to engage in a response, but the authors never test this hypothesis.

2. There are some inconsistencies and discrepancies in the data presentation which distract significantly from the points the authors attempt to make. Resolving inconsistencies would drastically improve the readability of the manuscript and increase the transparency of the data. For example:

a. The authors are inconsistent with their presentation of individual data points in some panels, but bars with only averages and SEM in others. These inconsistencies seem to be independent of both sample size and statistical test considerations. Additionally, they often occur within the same figure (Fig 1, 2, 4, 5, Suppl Fig 2). A specific example of this occurs with Fig 5h and Fig 5i, where sample sizes are reported to be the same, but one figure shows individual dot plots, while the other only

shows bars with SEM.

b. On page 5 line 22-23, the authors refer to titer data in Fig 1o which shows optical density data at an unspecified dilution rather than titer data. A comparison is then made to supplemental Fig 2e, which shows quantitative data making a true comparison difficult.

3. The results section of this manuscript is difficult to follow with most of the data tucked into the supplemental materials, much of which is key to understanding the logical flow of what is being presented. This frequently requires the reader to flip back and forth between the supplemental materials and the main text. While the excessive supplemental data is seemingly in response to the comments of prior reviewers, the manuscript would greatly benefit from a reworking of the goals of each figure and a reevaluation of which data are included. For example, in the sub-section titled "testosterone regulates the B-cell survival factor BAFF," the data that justifies looking at peripheral mechanisms, and thus BAFF, is tucked in supplemental Fig 3. Similarly, the data in support of the sub-section titled "AR mediated regulation of the splenic B-cell number is not intrinsic to B cells" is entirely located in the supplemental materials.

Minor:

1. The authors perform orchidectomies at both 4 weeks (pre-puberty) as well as 8 weeks (following sexual maturity). Comparisons are then made to females as well as male G-ARKO mice which are androgen receptor signaling deficient from early development. Given the differential roles that androgens, and androgen receptor signaling, play at different developmental time-points, the authors should address these differences and justify the variation in the timing of orchidectomy.

2. Supplemental Figure 1 should be omitted as it is an oversimplification of a complex pathway.

3. Refrain from referring to testosterone as a 'male hormone' as females have androgens as well. The difference is in the concentration.

Reviewers' comments:

Reviewer #1 (SLE, mouse models, BAFF)(Remarks to the Author):

This manuscript has already been reviewed by two expert reviewers and shows an interesting relationship between serum testosterone and BAFF levels that appears to be regulated by the neural reflex. While the physiologic significance of this finding is not yet known, the data shown in the manuscript is clear and the responses to the reviewers are detailed and in accordance with what is known about the physiology of BAFF with respect to B cells. The authors have clearly defined the scope of this manuscript and have appropriately answered those points within the paper's scope.

AUTHOR REPLY 2:

We thank reviewer #1 for his/her comments.

Reviewer #2 (Hormone/immune crosstalk)(Remarks to the Author):

The primary goal of this manuscript was to determine the direct effects of androgen receptor signaling on B cell activity, which could have implications for better understanding protection of males from autoimmune diseases, including rheumatoid arthritis. The authors show that the impact of androgen receptor signaling on B cells is mediated by B-cell activation factor (BAFF), which is a survival factor for B cells. Males have lower levels of BAFF than females and castration of males increases concentrations. Deletion of androgen receptors completely or specifically in osteoblast resulted in elevated levels of BAFF illustrating that androgen receptor signaling regulates BAFF, which in the spleen is associated with regulation of noradrenaline.

Reviewer comment

Major

1. Functional significance. From the previous review, the reviewers requested that the authors downplay the translational aspects of these studies. The fundamental problem with this paper is not that they need to downplay the functional significance, but rather that they have to show the functional significance for disease. They attempt this by measuring anti-bovine DNA responses and show that these antibodies are elevated in complete AR knock out mice. But then, in Suppl. Fig 2, they show that these animals have more B cells, but lower IgG antibody titers. To make their observed effects of AR signaling on BAFF significant, they would need to show that this is relevant to an antibody-mediated disease, such as rheumatoid arthritis, for which there are animal models. Without this, this observation lacks significance and with discrepant findings buried in the supplemental figures (being added from the previous review) it is difficult to determine if any of the reported effects would matter for a disease state. In its current state, the entire paper is a study of how AR signaling affects baseline B cells, which to be honest is often quite marginal. There may be larger effects if B cells were challenged to engage in a response, but the authors never test this hypothesis.

AUTHOR REPLY 3:

The main aim of this paper was to study the mechanism by which testosterone regulates splenic B-cell number in male mice and our main finding is reflected in the title of our paper. Castration is used in the paper to show main mechanisms of testosterone deficiency, for example blockade of the effect on B cell number by the BAFFR blocking antibody or the neurotoxin 6-OHDA. We suggest that regulation of BAFF and splenic B-cell number by testosterone is AR-dependent, but our aims and conclusions do not include how AR signaling affects disease models via BAFF.

The main endogenous AR ligands are testosterone and its metabolite dihydrotestosterone (DHT); the latter is mainly produced from testosterone locally in target tissues. While DHT mainly is a potent AR agonist, testosterone may exert some of its actions by aromatization to estradiol. For the purpose of targeting the functional significance of AR signaling on BAFF in animal models of antibody-mediated disease, possible approaches are considered and discussed below:

- Castration (testosterone/androgen deficiency). In the male mouse, removal of the testes equals removal of testosterone and thereby the solely most important endogenous androgen and AR ligand in male mice. Previous studies have performed castration studies in models of antibody-mediated diseases (see Table below).
- General AR knockout (G-ARKO). Due to developmental defects, G-ARKO males are completely testosterone deficient. Therefore, the effects of AR deficiency cannot be distinguished from those of testosterone deficiency in the G-ARKO model. In accordance, we find that phenotypes regarding for example serum BAFF levels and splenic B-cell number are similar in G-ARKO and castration models.
- Cell-specific ARKO model. The AR target cell for BAFF and splenic B-cell regulation remains unidentified, and it is therefore not yet possible to study the functional relevance of the AR signaling on BAFF in a cell-specific ARKO model.
- Testosterone treatment. Previous studies have performed testosterone treatment studies in models of antibody-mediated diseases (see Table below).
- DHT (endogenous AR agonist) treatment. Previous studies have performed DHT treatment studies in models of antibody-mediated diseases (see Table below).

Selected studies on castration and testosterone/AR agonist treatment in animal models of autoimmune disease.

Reference	Mouse model	Intervention	Main result
Roubinian et al. J Clin Invest 1977 ¹	NZB/NZW F1	Castration	Castration caused premature death in male mice
Roubinian et al. J Exp Med 1978 ²	NZB/NZW F1	Castration + AR agonist DHT	DHT improved survival, reduced anti-DNA ab, and reduced glomerulonephritis in male mice
Melez et al. J Immunopharmacol. 1978 ³	NZB/NZW F1	Castration + testosterone	Testosterone treatment of female mice reduced anti-DNA ab and increased survival
Roubinian et al. J Clin Invest 1979 ⁴	NZB/NZW F1	Castration + AR agonist DHT	DHT improved survival and reduced glomerulonephritis in female mice
Ganesan et al. Bone 2008 ⁵	Rat CIA model	Castration + AR agonist DHT	Castration worsened and DHT improved collagen-induced arthritis in male rats
Ichii et al. Lupus 2009 ⁶	B6.MRLc1(82- 100)	Castration	Castration increased glomerulonephritis in male mice
Keith et al. Arthritis Rheum 2013 ⁷	SKG	Castration	Castration increased arthritis, lung disease, and autoantibody generation in male mice

The rationale for not including a mouse model in the present paper is summarized in the following points:

- The scope of the paper was to study the mechanism by which testosterone regulates splenic B-cell number in male mice.
- We suggest that regulation of BAFF and splenic B-cell number by testosterone is AR-dependent, but our aims and conclusions do not include how AR signaling affects disease models via BAFF.
- The AR target cell for BAFF regulation remains unidentified. Therefore, while a cell-specific knockout of the AR may be useful future approach for the question how AR signaling affects disease models via BAFF, it is not an option for the present study.
- In the G-ARKO mouse model, effects of AR deficiency cannot be distinguished from those of testosterone deficiency.
- A number of published papers have shown effects of castration, testosterone treatment and AR agonist treatment in animal models of autoimmune disease.

To further clarify available data from disease models, a new paragraph has been added to the Discussion (page 13, line 22; cited under AUTHOR REPLY 4 below). We have also removed the phrase “.....providing a possible mechanism for the protection by androgens and for the sexual dimorphism in autoimmune disorders” from the first paragraph of the Discussion (page 11, line 4).

AUTHOR REPLY 4:

As the reviewer points out, total IgG levels were added in response to a previous reviewer comment, to exclude the possibility that increased anti-DNA IgG could be explained by generally increased IgG levels in G-ARKO mice. To increase transparency of data presentation, the total IgG data are now shown together with anti-DNA IgG in main Figure 1o, instead of in Supplementary information. See also clarification regarding the antibody data in AUTHOR REPLY 7. Further, while ANA antibody finding in G-ARKO mice was not at all discussed in the previous version of the paper, we have now added the following paragraph to the Discussion (page 13, line 22):

“While we suggest that testosterone regulates BAFF and splenic B-cell number in an AR-dependent manner, the functional relevance of this regulation for antibody-mediated diseases remains unclear. G-ARKO mice displayed increased anti-DNA antibodies that may be associated with autoimmune disease⁸, but we have not studied disease development in G-ARKO or castrated mice. However, castration has previously been shown to aggravate disease in different animal models of autoimmune disease, such as experimental lupus and arthritis models¹⁻⁷. Further, treatment with both testosterone and the endogenous AR agonist dihydrotestosterone ameliorates experimental lupus and arthritis²⁻⁵, supporting a protective effect of AR activation. Importantly, the role of BAFF for these beneficial effects of testosterone/AR agonist in experimental disease models remains unclear. Once the AR target cell for BAFF regulation is identified, a knockout of the AR specifically in this cell type may be a useful approach to address if AR signaling affects autoimmunity via BAFF regulation.”

Reviewer comment

2. There are some inconsistencies and discrepancies in the data presentation which distract significantly from the points the authors attempt to make. Resolving inconsistencies would drastically improve the readability of the manuscript and increase the transparency of the data.

AUTHOR REPLY 5:

We now have tried to resolve and/or clarify discrepancies in data presentation to improve the readability of the manuscript, see AUTHOR REPLY 6 below. See also changes detailed in AUTHOR REPLY 8.

For example:

a. The authors are inconsistent with their presentation of individual data points in some panels, but bars with only averages and SEM in others. These inconsistencies seem to be independent of both sample size and statistical test considerations. Additionally, they often occur within the same figure (Fig 1, 2, 4, 5, Suppl Fig 2). A specific example of this occurs with Fig 5h and Fig 5i, where sample sizes are reported to be the same, but one figure shows individual dot plots, while the other only shows bars with SEM.

AUTHOR REPLY 6:

We have chosen to present all flow cytometry data as individual data points, while other data are shown as means \pm SEM. This is now clarified in the figure legends of Figures 1, 3, 5 and 6 (previous Figures 1, 2, 4 and 5). Supplementary figure 2 has been revised as detailed in AUTHOR REPLY 8. If requested by the editor, all figure panels may be redrawn.

b. On page 5 line 22-23, the authors refer to titer data in Fig 1o which shows optical density data at an unspecified dilution rather than titer data. A comparison is then made to supplemental Fig 2e, which shows quantitative data making a true comparison difficult.

AUTHOR REPLY 7:

For increased transparency, data on total IgG levels in G-ARKO mice have been moved from Supplementary information to the same figure as the anti-DNA data (new main Figure 1o-p). For these measures, the serum was serially diluted and for total IgG we used 1:250000 and for anti-DNA we used 1:50; this is now clarified in Methods (page 18, line 20). The reason for using OD for the anti-DNA is that these are relatively low affinity antibodies and it is therefore difficult to find a good representative control serum or standard antibody.

Reviewer comment

3. The results section of this manuscript is difficult to follow with most of the data tucked into the supplemental materials, much of which is key to understanding the logical flow of what is being presented. This frequently requires the reader to flip back and forth between the supplemental materials and the main text. While the excessive supplemental data is seemingly in response to the comments of prior reviewers, the manuscript would greatly benefit from a reworking of the goals of each figure and a reevaluation of which data are included. For example, in the sub-section titled “testosterone regulates the B-cell survival factor BAFF,” the data that justifies looking at peripheral mechanisms, and thus BAFF, is tucked in supplemental Fig 3. Similarly, the data in support of the sub-section titled “AR mediated regulation of the splenic B-cell number is not intrinsic to B cells” is entirely located in the supplemental materials.

AUTHOR REPLY 8:

To increase the readability and clarity of the manuscript we have now moved certain data from supplementary to main figures. The following changes, with corresponding changes in figure legends, have been made:

- Data from former Supplementary Figure 3 is now included as a main figure (new Figure 2). The title of the new Figure 2 (“Splenic mature B cell number is not altered by AR depletion in osteoblasts”) now also appears in the Results section of the manuscript.
- Data from former Supplementary Table 2 is now included as a main table (new Table 1, entitled “AR-mediated regulation of splenic B-cell number is not intrinsic to B cells”).
- Data on serum BAFF levels in male and female mice have been moved from Supplementary information to the new main Figure 3c.
- Data on total IgG levels in G-ARKO mice have been moved from Supplementary information to the new main Figure 1o.

Reviewer comment

Minor:

1. The authors perform orchidectomies at both 4 weeks (pre-puberty) as well as 8 weeks (following sexual maturity). Comparisons are then made to females as well as male G-ARKO mice which are androgen receptor signaling deficient from early development. Given the differential roles that androgens, and androgen receptor signaling, play at different developmental time-points, the authors should address these differences and justify the variation in the timing of orchidectomy.

AUTHOR REPLY 9:

Castrations were performed in male mice aged 8-12 weeks throughout the paper, as we have learned that we get a full castration effect on splenic B cell number even after puberty. The single exception is the T cell depletion experiment (Supplementary Fig. 4), where the mice were castrated before puberty. Independently of pre- or post-puberty surgery, castration results in roughly a doubling of the number of splenic B cells. This is similar to our findings in G-ARKO mice in which the AR deletion is present from embryonic stage. Collectively, these observations do not support a crucial role for developmental or programming effects for the regulation of splenic B cell number by testosterone/AR. The following sentence now has been added to the Discussion (page 11, line 10):

“Short-term castration of adult mice resulted in a doubling of the number of splenic B cells, similarly to G-ARKO mice in which the AR deletion is present from embryonic stage⁹, suggesting that the regulation of splenic B-cell number by testosterone/AR occurs independently of developmental effects.”

Reviewer comment

2. Supplemental Figure 1 should be omitted as it is an oversimplification of a complex pathway.

AUTHOR REPLY 10:

This drawn figure illustrates our proposed mechanistic pathway and we agree that it is an oversimplification. If the editor concurs, we will remove the illustration from the Supplement.

Reviewer comment

3. Refrain from referring to testosterone as a 'male hormone' as females have androgens as well. The difference is in the concentration.

AUTHOR REPLY 11:

This sentence has now been adjusted (Introduction, page 3 line 3).

REFERENCES

1. Roubinian, J.R., Papoian, R. & Talal, N. Androgenic hormones modulate autoantibody responses and improve survival in murine lupus. *J Clin Invest* **59**, 1066-1070 (1977).
2. Roubinian, J.R., Talal, N., Greenspan, J.S., Goodman, J.R. & Siiteri, P.K. Effect of castration and sex hormone treatment on survival, anti-nucleic acid antibodies, and glomerulonephritis in NZB/NZW F1 mice. *J Exp Med* **147**, 1568-1583 (1978).
3. Melez, K.A., Reeves, J.P. & Steinberg, A.D. Regulation of the expression of autoimmunity in NZB x NZW F1 mice by sex hormones. *J Immunopharmacol* **1**, 27-42 (1978).
4. Roubinian, J.R., Talal, N., Greenspan, J.S., Goodman, J.R. & Siiteri, P.K. Delayed androgen treatment prolongs survival in murine lupus. *J Clin Invest* **63**, 902-911 (1979).
5. Ganesan, K., Balachandran, C., Manohar, B.M. & Puvanakrishnan, R. Comparative studies on the interplay of testosterone, estrogen and progesterone in collagen induced arthritis in rats. *Bone* **43**, 758-765 (2008).
6. Ichii, O., *et al.* Onset of autoimmune glomerulonephritis derived from the telomeric region of MRL-chromosome 1 is associated with the male sex hormone in mice. *Lupus* **18**, 491-500 (2009).
7. Keith, R.C., *et al.* Testosterone is protective in the sexually dimorphic development of arthritis and lung disease in SKG mice. *Arthritis Rheum* **65**, 1487-1493 (2013).
8. Suurmond, J. & Diamond, B. Autoantibodies in systemic autoimmune diseases: specificity and pathogenicity. *J Clin Invest* **125**, 2194-2202 (2015).
9. De Gendt, K., *et al.* A Sertoli cell-selective knockout of the androgen receptor causes spermatogenic arrest in meiosis. *Proc Natl Acad Sci U S A* **101**, 1327-1332 (2004).

REVIEWERS' COMMENTS:

Reviewer #2 (Remarks to the Author):

While the authors were attentive to reorganizing the figures and presentation of data, the fundamental problems with the study still remain.

1. Although it is true that "A number of published papers have shown effects of castration, testosterone treatment and AR agonist treatment in animal models of autoimmune disease", nobody has shown that the effects of androgen signaling on autoimmune diseases is caused by regulation of BAFF. This is what would make this paper novel and show functional significance and that is what this paper does not have. While the discussion section has been redone to broaden context, then introduction still sets the stage for considering androgen signaling in autoimmune disease. It is not too much to ask for the authors to determine if BAFF is unregulated in autoimmune disease mouse models, of which there are many.

2. There is still no explanation for why mice, that are not prone to autoimmune diseases, are making anti-DNA IgG antibodies. While this is a small attempt at functional significance, it is very confusing and not adequately explained. Are the authors insinuating that G-ARKO mice spontaneously develop autoimmune disease? Because if that is the case, then the authors are left with the task of conducting a detailed analysis of the pathogenesis of this disease state and not just merely measuring anti-DNA antibodies which are still out of context in the paper.

REVIEWERS' COMMENTS:

Reviewer #2 (Remarks to the Author):

While the authors were attentive to reorganizing the figures and presentation of data, the fundamental problems with the study still remain.

1. Although it is true that "A number of published papers have shown effects of castration, testosterone treatment and AR agonist treatment in animal models of autoimmune disease", nobody has shown that the effects of androgen signaling on autoimmune diseases is caused by regulation of BAFF. This is what would make this paper novel and show functional significance and that is what this paper does not have. While the discussion section has been redone to broaden context, then introduction still sets the stage for considering androgen signaling in autoimmune disease. It is not too much to ask for the authors to determine if BAFF is unregulated in autoimmune disease mouse models, of which there are many.

AUTHOR REPLY 1:

The Introduction has now been slightly revised, such that the starting sentence (page 4, line 1) is "Sex steroid hormones have profound effects on the immune system, and insight into these effects may provide important clues to the sexual dimorphism of immune-dependent disorders." We hope that this is a better opening for our topic on testosterone effects on the immune system. Nevertheless, we feel that the wider disease perspective is relevant to introduce.

The first paragraph of the discussion has been rewritten and references to human disease have been omitted from this paragraph (first paragraph on page 12).

Several papers show increased BAFF levels in experimental lupus models and also the efficacy of BAFF inhibition in these models. We now cite some of these references in the Discussion (page 14, line 23; references #52 and 53).

2. There is still no explanation for why mice, that are not prone to autoimmune diseases, are making anti-DNA IgG antibodies. While this is a small attempt at functional significance, it is very confusing and not adequately explained. Are the authors insinuating that G-ARKO mice spontaneously develop autoimmune disease? Because if that is the case, then the authors are left with the task of conducting a detailed analysis of the pathogenesis of this disease state and not just merely measuring anti-DNA antibodies which are still out of context in the paper.

AUTHOR REPLY 2:

Spontaneous development of anti-DNA antibodies is occurring in older mice (Nusser et al. Eur J Immunol 2014;44:2893) and is not specific to our models; we feel that trying to explaining this phenomenon goes beyond the format of this paper.

We agree that the functional significance of increased anti-DNA antibodies in G-ARKO mice relative to controls is unclear. This is emphasized in the Discussion (page 14, line 16):

"While we suggest that testosterone regulates BAFF and splenic B cell number in an AR-dependent manner, the functional relevance of this regulation for antibody-mediated diseases remains unclear. G-ARKO mice displayed increased anti-DNA antibodies that may be associated with autoimmune disease¹, but we have not studied disease development in G-ARKO or castrated mice. However, castration has previously been shown to aggravate disease in different animal models of autoimmune disease, such as experimental lupus and arthritis models²⁻⁸. Further, treatment with both testosterone and the endogenous AR agonist dihydrotestosterone ameliorates experimental lupus and arthritis³⁻⁶, supporting a protective effect of AR activation. Importantly, while BAFF inhibition may be protective in experimental lupus models^{9,10}, the importance of BAFF regulation for the beneficial effects of testosterone/AR agonist in experimental disease models remains unclear. Once the AR target cell for BAFF regulation is identified, a knockout of the AR specifically in this cell type may be a useful approach to address if AR signalling affects autoimmunity via BAFF regulation."

References

1. Suurmond, J. & Diamond, B. Autoantibodies in systemic autoimmune diseases: specificity and pathogenicity. *J Clin Invest* **125**, 2194-2202 (2015).
2. Roubinian, J.R., Papoian, R. & Talal, N. Androgenic hormones modulate autoantibody responses and improve survival in murine lupus. *J Clin Invest* **59**, 1066-1070 (1977).
3. Roubinian, J.R., Talal, N., Greenspan, J.S., Goodman, J.R. & Siiteri, P.K. Effect of castration and sex hormone treatment on survival, anti-nucleic acid antibodies, and glomerulonephritis in NZB/NZW F1 mice. *J Exp Med* **147**, 1568-1583 (1978).
4. Melez, K.A., Reeves, J.P. & Steinberg, A.D. Regulation of the expression of autoimmunity in NZB x NZW F1 mice by sex hormones. *J Immunopharmacol* **1**, 27-42 (1978).
5. Roubinian, J.R., Talal, N., Greenspan, J.S., Goodman, J.R. & Siiteri, P.K. Delayed androgen treatment prolongs survival in murine lupus. *J Clin Invest* **63**, 902-911 (1979).
6. Ganesan, K., Balachandran, C., Manohar, B.M. & Puvanakrishnan, R. Comparative studies on the interplay of testosterone, estrogen and progesterone in collagen induced arthritis in rats. *Bone* **43**, 758-765 (2008).
7. Ichii, O., *et al.* Onset of autoimmune glomerulonephritis derived from the telomeric region of MRL-chromosome 1 is associated with the male sex hormone in mice. *Lupus* **18**, 491-500 (2009).
8. Keith, R.C., *et al.* Testosterone is protective in the sexually dimorphic development of arthritis and lung disease in SKG mice. *Arthritis Rheum* **65**, 1487-1493 (2013).
9. Li, W., Titov, A.A. & Morel, L. An update on lupus animal models. *Curr Opin Rheumatol* **29**, 434-441 (2017).
10. Ding, H., *et al.* Blockade of B-cell-activating factor suppresses lupus-like syndrome in autoimmune BXSB mice. *J Cell Mol Med* **14**, 1717-1725 (2010).